# Comparison of Anti-SARS-CoV-2 IgG Antibody Responses Generated by the Administration of Ad26.COV2.S, AZD1222, BNT162b2, or CoronaVac: Longitudinal Prospective Cohort Study in the Colombian Population, 2021/2022

**DOI:** 10.3390/vaccines10101609

**Published:** 2022-09-26

**Authors:** Jeadran Malagón-Rojas, Marcela Mercado-Reyes, Yesith G. Toloza-Pérez, Marisol Galindo, Ruth M. Palma, Jenssy Catama, Juan F. Bedoya, Eliana L. Parra-Barrera, Ximena Meneses, Juliana Barbosa, Pilar Tavera-Rodríguez, Andrea Bermúdez-Forero, Martha Lucía Ospina-Martínez

**Affiliations:** 1Instituto Nacional de Salud, Bogotá 111321, Colombia or; 2PhD (C) Modelado en Política y Gestión Pública, Universidad Jorge Tadeo Lozano, Bogotá 111711, Colombia

**Keywords:** COVID-19, SARS-CoV-2, cohort, vaccine, Colombia

## Abstract

To mitigate the coronavirus disease 2019 (COVID-19) pandemic caused by severe acute respiratory syndrome coronavirus 2 (SARS-CoV-2), vaccines have been rapidly developed and introduced in many countries. In Colombia, the population was vaccinated with four vaccines. Therefore, this research aimed to determine the ability of the vaccines introduced in the National Vaccination Plan to prevent SARS-CoV-2 infection and induce seroconversion and sought to investigate the longevity of antibodies in the blood. We conducted a prospective, nonprobabilistic, consecutive cross-sectional cohort study in a population with access to vaccination with CoronaVac, Ad26.COV2.S, AZD1222, and BNT162b2 from March 2021 to March 2022. The study included 1327 vaccinated people. A plurality of participants were vaccinated with BNT162b2 (36.1%; *n* = 480), followed by Ad26.COV2.S (26.9%; *n* = 358), CoronaVac (24%; *n* = 331), and AZD1222 (11.9%; *n* = 158). The crude seroprevalence on day zero varied between 18.1% and 57.8%. Participants who received BNT162b2 had a lower risk of SARS-CoV-2 infection than those who received the other vaccines. Participants who were immunized with BNT162b2 and AZD1222 had a higher probability of losing reactivity on day 210 after receiving the vaccine.

## 1. Introduction

Vaccination is considered the primary strategy for preventing severe forms of coronavirus disease 2019 (COVID-19) [1]. By March 2022, approximately 64.3% of the world’s population had received at least one dose of a vaccine against COVID-19 [2]. According to the World Health Organization (WHO), 11,190 million doses have been administered worldwide, and 15.6 million are now administered daily [2]. Evaluating the effectiveness of the COVID-19 vaccines available in the different populations where the vaccines have been administered is necessary for the generation of knowledge and strategies for both current and future vaccination plans.

Currently, there are approximately 336 studies in progress evaluating different vaccines in different populations and regions of the world [3,4]. The generation of this type of technology, as well as greater flexibility in the requirements for approving research in humans, has led to doubts among a large portion of the population [5,6]. Additionally, there is variation in vaccine efficacy for severe forms of the disease [7]. In turn, concerns have been raised about the representation of indigenous and Hispanic populations within clinical trials for vaccines, a lack of which could potentially impact seroconversion outcomes [8,9].

Several countries in the region have provided reports on the efficacy of vaccines used in their populations. For example, Chile reported that the effectiveness of the CoronaVac vaccine was 65.9% (95% CI, 65.22–66.6%) for the prevention of the disease, 87.5% (95% CI, 86.7–88.2%) for hospitalization, and 86.3% (95% CI, 84.5–87.9%) for avoiding deaths related to COVID-19 [10]. In Uruguay, the effectiveness of the BNT162b2 vaccine was 78% (95% CI, 76–79%), and for CoronaVac, it was 59.9% (95% CI, 59–60%) [11]. In turn, the efficacy in preventing deaths was 96% (95% CI, 95.3–96.8%) for BNT162b2 and 94.65% (95% CI, 93.4–95.6%) for CoronaVac. Additionally, a review of the literature reported that full vaccination with COVID-19 vaccines is highly effective against early COVID-19 variants (alpha, beta, gamma, and delta), whereas the effectiveness against the delta and omicron variants is limited [12].

The variation in the studies carried out in different regions suggests that there are elements at the individual level that influence the outcome evaluation.

An additional aspect that has not been addressed in vaccine efficacy studies is the effect that they have on the immune system. Although health systems need to understand the effect of vaccination schedules on death and transmission, it is also relevant to determine the effect that vaccines have on the immune system at the population level, including humoral and cell responses [13]. This information is generally of interest only for the first phases of clinical trials, but in the context of emerging new strains of interest, with potentially greater transmission or lethality, it is important to fully understand how vaccines stimulate effector cells and immunoglobulin production [14,15] Recently, concerns have arisen about the need for administering additional doses or combinations of vaccines to ensure long-term immunity. A better understanding of the effect that vaccination schedules have on the real-world population may inform decision-making and potential modifications to the National Vaccination Plan. Colombia implemented the National Vaccination Plan against severe acute respiratory syndrome coronavirus 2 (SARS-CoV-2), in which vaccination schedules were established for WHO-approved vaccines acquired through multilateral mechanisms (COVAX) and by direct negotiation with pharmaceutical companies. The aim of this study was to evaluate the effect of vaccines available in the National Vaccination Plan on the immune system of the Colombian population through the prospective measurement of total antibody markers.

## 2. Materials and Methods

### 2.1. Type of Study

This was a nonprobabilistic, consecutive cross-sectional study of a Colombian population who received a vaccination schedule with Ad26.COV2.S (Janssen, Beerse, Belgium), AZD1222 (AstraZeneca, Cambridge, UK), BNT162b2 (Pfizer, New York, NY, USA), or CoronaVac (Sinovac, Beijing, China) through the National Vaccination Plan in the prioritization phase established by the National Government.

### 2.2. Timeline

The National Vaccination Plan established specific criteria for the population and the type of vaccine used based on the availability of resources. In this sense, for the study population, the BNT162b2 vaccine was available to the population between March and June 2021; the CoronaVac vaccine was available between May and August 2021; and AZD1222 and Ad26.COV2 were available between June and July 2021. This study was conducted between March 2021 and March 2022.

### 2.3. Eligibility Criteria

Men and women between 18 and 80 years old who were residents of the Colombian territory were included. Additionally, volunteers who, at the time of inclusion in the study, intended to receive the complete vaccination schedule for COVID-19 were enrolled.

Participants with comorbidities such as diabetes, cardiovascular diseases, hypo/hyperthyroidism, chronic kidney disease, or some type of physical disability were excluded. Additionally, participants who, due to time availability, did not participate in the scheduled follow-ups and who had medical contraindications were excluded from the study.

### 2.4. Population

The global sample size was estimated for the entire study, using the percentage of seroconversion generated after the administration of the vaccination schedules as a parameter [16]. The sample size was estimated by using OpenEpi v 3.0. A sample size of 589 participants was estimated, with a confidence level of 95%, a margin of error of 5%, an accuracy of 2.5%, and an oversample of 30%. Subsequently, proportional allocation was employed to assign participants to groups for each vaccine proposed in the National Vaccination Plan [17].

Participants were recruited from five health institutions: *Santa Matilde* Hospital in Madrid; *Santa Matilde* Hospital—Bojacá; National Institute of Health; *San Andrés* Island Departmental Laboratory of Public Health; and *San Rafael de Fusagasugá* Hospital. After the vaccine was administered, individuals were invited to participate in the study. The people who voluntarily agreed to participate were directed to a site where information was collected.

### 2.5. Sociodemographic Characterization and Adverse Reactions

Each participant responded to a survey designed to collect information regarding health conditions, sociodemographic variables, and symptoms associated with vaccination.

We considered adverse reactions reported by the National Health Services and WHO [18,19,20], including mild-to-moderate symptoms (pain at the injection site, fever, fatigue, headache, muscle pain, chills, and diarrhea) and severe effects (anaphylaxis, thrombosis with thrombocytopenia syndrome, Guillain-Barré syndrome, myocarditis, pericarditis, and death).

### 2.6. Biological Sample Collection

A blood sample (7 mL) was collected from each participant by venipuncture, using additive-free silicone vacuum tubes containing serum separator gel. Subsequently, the samples were centrifuged at 3500 rpm for 10 min. Then 2 mL of serum was stored in vials and frozen at −70 °C until processing.

Nasopharyngeal samples were collected by using sterile nylon microbiological transport swabs (Improve^®^, Guangzhou, China) in a viral transport medium (VTM) (Viral ad-bio^®^) and frozen at −70 °C until processing.

### 2.7. RNA Identification

Nasopharyngeal swab samples were refrigerated and transported to the local laboratory in each city. The RNA was extracted and amplified according to the Berlin protocol, which was validated by the Insituto Nacional de Salud [21]. Viral RNA extraction was performed by using a QIAamp Viral RNA Mini Kit (Qiagen Inc., Chatsworth, CA, USA) or the MagNA Pure LC nucleic acid extraction system (Roche Diagnostics GmbH, Mannheim, Germany) [21].

### 2.8. Antibody Identification

The ADVIA Centaur COV2 (COV2T and sCOV2G) assay was used to measure total antibodies (IgM + IgG). The COV2T is an in-sandwich one-step automated antigen test for the qualitative detection of total antibodies (IgM + IgG) against the receptor-binding domain (RBD) of the spike protein of SARS-CoV-2 in serum or plasma. The ADVIA Centaur COV2 is a fully automated antigen sandwich immunoassay, using acridinium ester chemiluminescent technology, in which antigens are bridged by antibodies present in the sample. The solid phase contains a preformed complex of streptavidin-coated microparticles and biotinylated SARS-CoV-2 spike 1 receptor-binding domain (S1 RBD) recombinant antigens. This reagent is used to capture anti-SARS-CoV-2 antibodies in the sample [21]. The Lite Reagent contains acridinium-ester-labeled SARS-CoV-2 S1 RBD recombinant antigens used to detect anti-SARS-CoV-2 antibodies bound to the solid phase.

### 2.9. Follow-Up

The members of our team (authors and collaborators) made telephone calls to remind participants of collection dates and to deliver results. The telephone calls were made two days before the date of the second dose and on day 28, day 58, day 88, and day 209.

The follow-up times for the AZD1222 vaccine were days 0, 84, 110, 144, and 210 after the first dose; for BNT162b2, the follow-up times were days 0, 21, 60, 90, and 210 after the first dose; and for CoronaVac and Ad26.COV2.S, the follow-up times were days 0, 30, 60, 90, and 210 after the first vaccine dose.

### 2.10. Statistical Analysis

The absolute and relative frequencies are presented for qualitative variables. For quantitative variables, the results are presented as averages and standard deviations. The assumption of normality was validated by using the Shapiro–Wilk test. Subsequently, a comparison between categorical variables was performed by using the chi-square test or Fisher′s exact test.

To evaluate associations between the independent variables, a multivariate analysis model (*Poisson*) was applied that considered infection during the study time period and seroconversion at 210 days as the dependent variables. Variables with *p*-values < 0.2 in the bivariate analysis were included in the model. We adjusted the model for covariates to reduce bias in estimating the vaccination effect [22]. The most parsimonious model was selected based on the AIC score. The prevalence ratio (PR) was estimated with its respective 95% confidence interval.

The overall crude frequencies of seropositivity tests were estimated. Then crude seroprevalence was adjusted by using the Bayesian method in R V.2.21.2 (package RStan), using the sensitivity and specificity data reported in previous studies carried out by using a chemiluminescence immunoassay (CLIA) in Colombian populations [23]. The 95% Bayesian confidence intervals were obtained. The analyses were performed in R^®^ version 3.6.2.

## 3. Results

### 3.1. Sociodemographic Characteristics, Vaccination Schedules, and Clinical History

A total of 1327 vaccinated individuals were included. The majority (73.3%) had three or more follow-ups. The average age of all participants was 46.3 ± 15.85 years. On average, participants performed 15 ± 7.2 min of physical activity per day for five days a week. The average body mass index was 25 ± 5.8. It was found that 27.8% of the participants had at least one underlying pathology (*n* = 370) (95% CI, 25.4–30.3%); the most frequent was hypertension (13.4%; *n* = 185), followed by diabetes (4.1%; *n* = 54) (Table 1).

The majority of participants was vaccinated with BNT162b2 (36.1%; *n* = 480) and Ad26.COV2.S (26.9%; *n* = 358), followed by CoronaVac (24%; *n* = 331) and AZD1222 (11, 9%; *n* = 158) (Table 2). Participants who received BNT162b2 had a higher education level than those who received AZD1222 or Ad26.COV2.S (*p* > 0.005). Likewise, participants who received Ad26.COV2.S belonged to a lower socioeconomic stratum than those who received BNT162b2 (*p* < 0.005). Additionally, people who received AZD1222 were significantly older than those who received BNT162b2, CoronaVac, or Ad26.COV2.S (*p* < 0.005).

Eight percent (*n* = 101) of the participants had received a booster dose on day 210. Only participants who were vaccinated with Ad26.COV2.S (12%; *n* = 43) and CoronaVac (17.7%; *n* = 58) received a booster. The average time between the first dose and the booster was 49 ± 7 days.

The loss proportion at the fourth follow-up was 44% (*n* = 589). Losses were more frequent in the groups who received Ad26.COV2.S (18.69%; *n* = 248) and CoronaVac (14.02%; *n* = 186); Figure 1 shows a flow diagram of the participants in the study.

### 3.2. Adverse Effects

The most common symptoms were pain at the injection site (64%), muscle pain (34%), chills (12%), and fever (8%). These symptoms existed for up to six days after receiving the vaccine. The participants recovered after 3 ± 1.5 days. Acute hypersensitivity and severe adverse effects were not reported.

### 3.3. Incidence of SARS-CoV-2

A total of 59.68% (*n* = 792) of the participants who did not have prior SARS-CoV-2 infection were included in the study (this was verified by using records obtained from the National Sampling System—SISMUESTRAS and ADVIA Centaur COV2).

A total of 10.22% of individuals with COVID-19 (95% CI, 8.25–12.49%; *n* = 81) were infected with SARS-CoV-2 at some point in the study. Most of the participants who contracted the infection during the study had asymptomatic (29.62%) and mild manifestations (70.37%). None of the infected individuals required inpatient management or supplemental oxygen. One of the participants who received AZD1222 passed away due to an acute myocardial infarction. Additionally, compared with male participants, female participants had a doubled risk of COVID-19 (Table 2). Participants who received BNT162b2 or AZD1222 were less likely to be infected than other participants. In turn, people who received Ad26.COV2.S or CoronaVac had a higher risk of SARS-CoV-2 infection (Table 2).

### 3.4. Seroprevalence

The crude seroprevalence for SARS-CoV-2 at baseline was 36.2% (95% CI, 33.5–38.9%). Ten percent (95% CI, 8.2–12.6%) of participants had a COVID-19 diagnosis during follow-ups, but none required hospitalization.

The crude seroprevalence on day zero varied between 18.1% and 57.8% (Figure 2). An association was found between being over 60 years old and CLIA reactivity at the time of inclusion in the study (RR, 0.55 (95%CI, 0.39–0.78)). The seroprevalence at the end of the study was greater than 70%.

Administration of the Ad26.COV2.S vaccine resulted in a crude seroprevalence of 57.8% (95% CI, 51.1–64.5%) on day zero, 93.1% (95% CI, 88–97%) on day 30, and 91.1% (95% CI, 87–95.2%) on day 60. After 60 days, the crude seroprevalence was 89.8% (95% CI, 85–94.7%); on day 90, the crude seroprevalence was 86.6% (95% CI, 81.4–91.9%); and on day 210, the crude seroprevalence was 97.4% (95% CI, 94.4–100.3%).

The crude seroprevalence for the AZD1222 vaccine on day zero was 18.1% (95% CI, 3.8–32.3%); on day 84, the crude seroprevalence was 69.9% (95% CI, 60.2–79.6%); on day 110, the crude seroprevalence was 96.5% (95% CI, 92.5–100.5%); on day 144, the crude seroprevalence was 93.5% (95% CI, 87.8–99.2%); and on day 210, the crude seroprevalence was 87.8% (95% CI, 79.9–95.8%).

For the CoronaVac vaccine, the crude seroprevalence on day zero was 45.2% (95% CI, 36.2–54.2%); on day 30, the crude seroprevalence was 69.2% (95% CI, 60.5–77.9%); on day 60, the crude seroprevalence was 93% (95% CI, 89.5–96.6%); on day 90, the crude seroprevalence was 88.9% (95% CI, 84.8–93%); and on day 210, the crude seroprevalence was 69.7% (95% CI, 60.7–78.6%).

At 21 days, 50.5% (95% CI, 46.9–54.3%) of the participants who were initially seronegative had developed antibodies (Figure 2); at day 210, the proportion of people who seroconverted was maintained; therefore, 6.45% (95% CI, 4.7–8.6%) of participants had not developed detectable IgG antibodies at study follow-up.

An association was found between seroprevalence (*p* < 0.001) and socioeconomic status, level of education, history of presenting with the disease, number of people living with or with a close relative positive for COVID-19, and age. From day zero to day 30, 3 people went from being reactive to being nonreactive, remaining nonreactive throughout follow-up; from day 30 to day 60, 4 people went from reactive to nonreactive, remaining nonreactive the rest of the study; from day 60 to day 90, 16 people went from being reactive to nonreactive; and from day 90 to day 210, 36 people went from being reactive to being nonreactive.

Figure 2 shows the proportion of participants who were reactive during the follow-up time (crude in blue and adjusted seroprevalence in red). The crude proportion shows the proportion of the participants who were reactive. To determine the real proportion of seroreactivity, an adjustment was performed based on the sensitivity and specificity of the ADVIA Centaur COV2, as previously reported [23]. The figure also presents the prevalence of total antibodies against COVID-19 during the study (days 0, 30, 60, 90, and 210). Participants who received BNT162b2 or AZD1222 had greater seroreactivity ratios than those who received Ad26.COV2 or CoronaVac. However, by day 210 of follow-up, these reactivity ratios were greater than 70% for participants administered the four vaccines.

### 3.5. Changes in Reactivity between Day 0 and Day 210

Among the participants who had a reactive test, 2.71% (95% CI, 1.84–3.59%; *n* = 36) had a nonreactive result on day 210.

When considering the factors associated with the loss of reactivity evaluated by CLIA, the participants vaccinated with CoronaVac or Ad26.COV2.S were 3.8 and 7.6 times more likely to become negative at day 210 than participants administered BNT162b2, respectively. Likewise, participants aged 60 years or older were five times more likely to cease being reactive than those younger than 60 years of age.

Participants who received CoronaVac or Ad26.COV2 schedules were 3.3 and 4.4 times more likely to become negative than participants who received BNT162b2. Likewise, participants older than 60 years were more likely to have a nonreactive test on day 210 of follow-up.

## 4. Discussion

In general, approved vaccines against COVID-19 have shown positive safety and efficacy profiles [24]. In this study, the vaccines administered were effective in preventing death and hospitalization associated with SARS-CoV-2 infection. Among the study participants, the SARS-CoV-2 infection rate was 10.22%. However, most of the population (>70%) remained positive for serum IgG antibodies until day 210 of follow-up.

Notably, when the participants who received BNT162b2 were recruited, the country was in a valley between the second and third peaks of infections. However, volunteers who received other vaccines were recruited when Colombia was going through the third wave of COVID-19, with active mu and delta variant circulation [25]. This context has implications for the analysis considering that participants who received BNT162b2 were thus less likely to be infected during the study period because the *Rt* for the March–August 2021 period was lower than the *Rt* for the June–December 2021 period (0.87 and 0.89, respectively) [26].

During the study period, no deaths or hospitalizations related to SARS-CoV-2 infection were reported. The incidence of SARS-CoV-2 infection was 10%, with all cases involving mild manifestations without self-reported side effects. However, the study participants who received Ad26.COV2.S had a higher risk of infection during the study period than those who received BNT162b2. This may be related to aspects not directly involved with immunity but rather to individual behaviors related to SARS-CoV-2 infection, such as socioeconomic status and education level. A national survey of the seroprevalence of SARS-CoV-2 in Colombia in 2020 found that people from lower socioeconomic strata and with lower education levels had a higher risk of having reactive antibodies against SARS-CoV-2 [21].

In the multivariate model, when controlling for social stratum and education level, people who received Ad26.COV2.S were twice as likely to be infected as those who received BNT162b2 were. This association was not observed with the other vaccines that required two doses. A randomized clinical trial that included 39,185 participants aimed to evaluate the effectiveness of Ad26.COV2.S and found that, in 52.9% of the participants, the vaccine was effective in preventing severe forms of COVID-19 and hospitalization owing to COVID-19 [27]. The same study found that the protection against the severe forms of infection generated by the delta and mu variants was much lower than that reported for the alpha, beta, and gamma variants.

In this sense, it is very likely that people who received Ad26.COV2.S needed a booster dose within one year, as originally suggested by the manufacturer. Overall, it has been reported that booster doses, both homologous and heterologous boosters, increase the levels of specific binding antibodies against protein S and neutralizing antibodies and increase T-cell responses but that these increases are greater in participants who received heterologous schedules with COVID-19 mRNA-based vaccines [28]. In our study, it was not possible to evaluate the effect of booster doses because this measure was not adopted in Colombia until November 2021, and by the time the cohort had completed follow-up, only 8% (*n* = 109) had received their booster dose. In the multivariate analysis, this variable did not influence the outcomes evaluated.

Regarding reactivity at day 210, people who received Ad26.COV2 or CoronaVac were more likely to have a nonreactive antibody test at day 210. Except for age, no individual variables were associated with this outcome. These findings coincide with previous results. A prospective cohort study with 384 participants in Cyprus evaluated the antibody response to the administration of BNT162b2, AZD1222, or CoronaVac and found that participants who received BNT162b2 had the highest seroreactivity ratios, followed by those who received AZD1222 [29]. Likewise, the authors reported that there was a greater proportion of participants with decreased reactivity among those who received CoronaVac, followed by those who received AZD1222 [29]. Furthermore, the authors found that the seroreactivity rate was lower among the participants who received AZD1222 (90%) or CoronaVac (60%) [29].

Another study conducted on health workers who were administered BNT162b2 found that participants who had been infected with SARS-CoV-2 (either before the study or during the study) and who completed their vaccination schedule maintained reactivity for longer than those who had received only the vaccine [30].

In summary, the administration of BNT162b2 seems to result in a greater immunogenic response among healthy adults. Some reports in the literature have reported that, in the absence of an accessible second-generation vaccine, heterologous boosting using BNT162b2 for inactivated and vectored primary vaccination recipients is preferred [31].

Our study has several limitations. First, there were limitations related to the nonprobabilistic consecutive sampling design. Due to the nature of this design, the conclusions are only applicable to this group of participants and cannot be extrapolated to other regions of the country. Second, due to the absence of a control group, it was not possible to compare the outcomes evaluated against the unvaccinated population. As a strength, this study was in agreement with a robust core of studies that have shown the effectiveness of vaccines against COVID-19, compared to a placebo, in reducing mortality and ICU hospitalization [10,32,33,34].

Third, this study evaluated only the total antibody and IgG responses and did not evaluate neutralizing antibodies or the cellular response against the virus. Recent reports carried out with serum from the Colombian population suggest a reduction in the ability of antibodies to neutralize the variants identified in 2021 compared to the ability of antibodies to neutralize the variants identified in 2020 and the original strain identified in 2019 [35].

Fourth, our study did not analyze the effects on the cellular response. An evaluation of the humoral response to vaccines is needed to understand how they enhance the immune response. Some studies have indicated that the administration of two doses of the BNT162b2 mRNA COVID-19 vaccine increases the expansion of antigen-specific CD4 + CD40L + T cells [36]. Additionally, it has been proposed that CoronaVac administration increases the levels of IFN-gamma- and granzyme B-producing cells [37].

Finally, although the study recruited a greater number of participants than the minimum sample size, 44% of participants who received Ad26.COV2 or CoronaVac did not complete the study. This can be explained based on two main reasons. Ad26.COV2 vaccination requires only one dose, which facilitated adherence to the study because there was no second “forced” encounter. In turn, for participants who received CoronaVac, the second dose was not administered until day 56 due to the lack of available vaccines in the country. With this, the second “forced” encounter was delayed, and in that process, a considerable number of participants were lost. Furthermore, the economic situation in the country forced many of the participants to drop out because they did not have permission from their employers to take time off from work to provide samples. However, because we had the identification numbers of the participants, we searched the death-record database to determine whether individuals had died.

## 5. Conclusions

Vaccines administered in the current schedule have shown a notable effect on the development of severe clinical manifestations. In addition, participants who received BNT162b2 or AZD1222 seem to have prolonged IgG reactivity.

Future studies should investigate how immunity is affected by different booster schedules, as well as assess neutralizing antibody titers for different variants. This research can be further extended to different vaccines and booster schedules that are currently in use.

## Figures and Tables

**Figure 1 vaccines-10-01609-f001:**
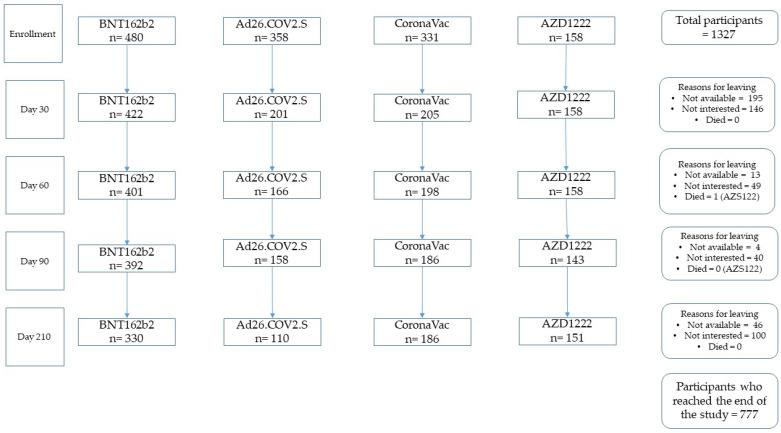
Flow diagram of the participants in the study.

**Figure 2 vaccines-10-01609-f002:**
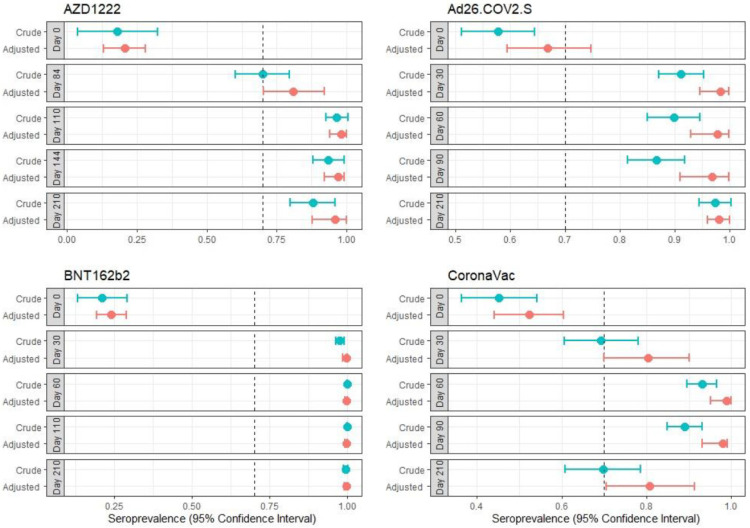
Crude and adjusted seroprevalence throughout the study.

**Table 1 vaccines-10-01609-t001:** Sociodemographic and clinical history of the cohort participants, 2021/2022.

			COVID-19 Prior to the Enrollment?							
			Yes % (*n*)				No % (*n*)			
Variable		Total (*n*)	AZD1222	Ad26.COV2. S	BNT162b2	CoronaVac	AZD1222	Ad26.COV2. S	BNT162b2	CoronaVac
Sex	Male	461	8.89 (16)	41.67 (75)	22.78 (41)	26.67 (48)	16.67 (45)	22.59 (61)	32.59 (88)	28.15 (76)
	Female	844	9.12 (29)	40.57 (129)	27.67 (88)	22.64 (72)	12.93 (68)	16.73 (88)	47.91 (252)	22.43 (118)
Age group	18–26	98	-	18.42 (7)	23.68 (9)	57.89 (22)	-	21.67 (13)	31.67 (19)	46.67 (28)
	27–59	916	0.26 (1)	46.56 (176)	29.89 (113)	23.28 (88)	0.37 (2)	24.35 (131)	56.32 (303)	18.96 (102)
	>60	313	49.44 (44)	24.72 (22)	7.87 (7)	17.98 (16)	49.55 (111)	4.02 (9)	12.95 (29)	33.48 (75)
Blood type	A	349	7.74 (12)	39.35 (61)	26.45 (41)	26.45 (41)	15.43 (27)	18.86 (33)	38.29 (67)	27.43 (48)
	AB	23	12.5 (1)	12.5 (1)	50 (4)	25 (2)	13.33 (2)	20 (3)	33.33 (5)	33.33 (5)
	B	116	6.67 (2)	36.67 (11)	33.33 (10)	23.33 (7)	9.3 (8)	10.47 (9)	47.67 (41)	32.56 (28)
	O	817	9.84 (30)	42.95 (131)	24.26 (74)	22.95 (70)	14.84 (76)	20.31 (104)	42.77 (219)	22.07 (113)
	Without data	22	-	14.29 (1)	-	85.71 (6)	-	-	-	100 (11)
Socioeconomic stratum	Low	601	10.79 (30)	43.17 (120)	12.95 (36)	33.09 (92)	19.81 (64)	29.41 (95)	21.98 (71)	28.79 (93)
	Medium	642	6.76 (14)	37.68 (78)	44.44 (92)	11.11 (23)	11.26 (49)	11.95 (52)	60.69 (264)	16.09 (70)
	High	47	-	-	50 (1)	50 (1)	-	-	36.36 (16)	63.64 (28)
	Without data	37	5.88 (1)	41.18 (7)	0 ()	52.94 (9)	-	30 (6)	-	70 (14)
Educational level	None	7	-	66.67 (2)	-	33.33 (1)	-	75 (3)	-	25 (1)
	Primary	166	21.74 (20)	61.9 6(57)	-	16.3 (15)	80.77 (42)	-	-	19.23 (10)
	High school	373	11.41 (21)	50 (92)	9.78 (18)	28.8 (53)	27.13 (51)	26.6 (50)	12.77 (24)	33.51 (63)
	College	225	1.12 (1)	32.58 (29)	31.46 (28)	34.83 (31)	2.21 (3)	25 (34)	32.35 (44)	40.44 (55)
	University	205	3.51 (2)	36.84 (21)	33.33 (19)	26.32 (15)	6.76 (10)	21.62 (32)	40.54 (60)	31.08 (46)
	Post-graduate	322	-	4.23 (3)	90.14 (64)	5.63 (4)	1.21 (3)	3.24 (8)	90.28 (223)	5.26 (13)
	Without data	29	11.11 (1)	11.11 (1)	-	77.78 (7)	20 (4)	20 (4)	-	60 (12)

**Table 2 vaccines-10-01609-t002:** Multivariate model of risk factors for COVID-19 during the study (*n* = 703).

Variable		*p*	PR (CI 0.95) (Unadjusted)	*p*	PR (CI 0.95) (Adjusted)
Sex	Female	0.018	2.343 (1.208–5.003)	0.033	2.182 (1.109–4.712)
	Male	1		1	
Age group	>60	0.024	0.368 (0.139–0.811)	0.730	1.188 (0.397–2.922)
	<60	1		1	
	Ad26.COV2.S	0.059	0.399 (0.135–0.947)	0.076	0.420 (0.142–1.001)
	AZD1222	0.008	0.068 (0.004–0.317)	0.012	0.062 (0.003–0.392)
	BNT162b2	1		1	
	CoronaVac	0.155	0.548 (0.220–1.183)	0.814	1.293 (0.068–7.750)
Socioeconomic level	High	1			
	Low	0.459	0.719 (0.270–1.601)		
Booster	Yes	0.678	0.10 (0.06–0.17)		
	No	1			

## Data Availability

Data are available upon reasonable request. Data may be obtained from a third party and are not publicly available.

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
