# Peer review of "Comparison of Anti-SARS-CoV-2 IgG Antibody Responses Generated by the Administration of Ad26.COV2.S, AZD1222, BNT162b2, or CoronaVac: Longitudinal Prospective Cohort Study in the Colombian Population, 2021/2022"

_vaccines, 2022, doi:10.3390/vaccines10101609_

Round 1
Reviewer 1 Report
This manuscript is a comparatively innovative investigative study, and its conclusions are illuminating. Below are some comments/suggestions in order to improve the MS.
(1) The figure legends and models in the paper are very vague, which makes it difficult for readers to quickly grasp the detailed meaning of the figures and the specific contents of the model. I would suggest the authors improve these parts to make the manuscript more readable.
(2) The authors can add more charts to show the analysis of different types of experimenters. The current analysis is not detailed enough.
(3) There are some deficiencies in the experimental design. The author highlighted the importance of exploring more about immunoassays in the introduction section, but no more related experiments were carried out. In addition, the effects of booster doses and unfinished experimenters were not strictly statistically analyzed, which needs to be supplemented. Another basic point is that the authors did not set up a control group that was not vaccinated, which is strange.
Author Response
Comment 1: This manuscript is a comparatively innovative investigative study, and its conclusions are illuminating. Below are some comments/suggestions in order to improve the MS. The figure legends and models in the paper are very vague, which makes it difficult for readers to quickly grasp the detailed meaning of the figures and the specific contents of the model. I would suggest the authors improve these parts to make the manuscript more readable.
Response to reviewer: Thank you so much for your comments, it really improves the reader´s understanding of the manuscript. We have corrected the figures and legends.
Comment 2: The authors can add more charts to show the analysis of different types of experimenters. The current analysis is not detailed enough.
Response to reviewer: Thanks for your suggestion. According to this, we have added an additional figure in the document to explain more about the types of experiments and the analysis. Also, we have improved the methods and results section, adding more details to the experiments performed. The changes are presented as follows:
Eligibility criteria
Men and women between 18 and 80 years old, who were residents of the Colombian territory. Also, voluntaries who at the time of inclusion in the study, intended to receive the complete vaccination schedule for COVID-19 were included.
The participants with these comorbidities were also excluded. We have changed this part of the document as follows:
In the analysis were exclude the participants with comorbidities such as diabetes, cardiovascular diseases, hypo/hyperthyroidism, chronic kidney disease, and living with some type of physical disability. Also, participants who, due to time availability, did not perform the scheduled follow-ups and who had medical contraindications were excluded from receiving biologics.
Participants who, due to time availability, did not perform the scheduled follow-ups and who had medical contraindications were excluded from the study.
Population
The global sample size was estimated for the entire study using as a parameter the percentage of seroconversion generated after the administration of the vaccination schedules [16]. The sample size was estimated using OpenEpi v 3.0. A sample size of 589 par-tic pants was estimated, with a confidence level of 95%, a margin of error of 5%, an accuracy of 2.5%, and an oversample of 30%. Subsequently, the proportional allocation was employed for assigning participants to groups for each vaccine proposed in the National Vaccination Plan [17].
Participants were recruited from five health institutions: Santa Matilde Hospital in Madrid, Santa Matilde Hospital - Bojacá, National Institute of Health, San Andrés Island Departmental Laboratory of Public Health, and San Rafael de Fusagasugá Hospital. After the vaccine was administered, individuals were invited to participate in the study. The people who voluntarily agreed to participate were directed to a site where information was collected.
Sociodemographic characterization and adverse reactions
Each participant responded to a survey designed to collect information regarding health conditions, sociodemographic variables, and symptoms associated with vaccination.
We considered adverse reactions reported by the National Health Services and World Health Organization [18–20]. It was considered mild-moderate symptoms (pain at the injection site, fever, fatigue, headache, muscle pain, chills and diarrhea) and severe effects (anaphylaxis, thrombosis with thrombocytopenia syndrome, Guillain-Barré Syndrome, myocaditis, pericarditis and death).
Biological samples collection
A blood sample (7 mL) was collected from each participant by venipuncture using additive-free silicone vacuum tubes containing serum separator gel. Subsequently, the samples were centrifuged at 3,500 rpm for 10 minutes. Then, 2 ml of serum was stored in vials and frozen at -70 °C until processing.
Nasopharyngeal samples were collected using sterile nylon microbiological transport swabs (Improve®, China) in viral transport medium (VTM) (Viral ad-bio®) and frozen at -70 °C until processing.
RNA identification
Nasopharyngeal swab samples were refrigerated and transported to the local laboratory in each city. The RNA was extracted and amplified according to the Berlin protocol, which was validated Instituto Nacional de Salud [21]. Viral RNA extraction was per-formed by using the QIAamp Viral RNA Mini kit (Qiagen Inc., Chatsworth, CA, USA) or the MagNA Pure LC nucleic acid extraction system (Roche Diagnostics GmbH, Mann-heim, Germany) [21].
Antibodies identification
The ADVIA Centaur COV2 (COV2T & sCOV2G) assay was used in order to measuring total antibodies (IgM+ IgG). The COV2T is an in-sandwich one-step automated antigen test for the qualitative detection of total antibodies (IgM + IgG), against the receptor-binding domain (RBD) of the spike protein of SARS-CoV-2 virus, in serum or plasma. The ADVIA CentaurCOV2 is a fully automated step antigen sandwich immunoassay using acridinium ester chemiluminescent technology, in which antigens are bridged by antibodies present in the sample. The Solid Phase contains a preformed complex of streptavidin-coated microparticles and biotinylated SARS-CoV-2 spike 1 receptor binding domain (S1 RBD) recombinant antigens. This reagent is used to capture anti-SARS-CoV-2 antibodies in the sample [21]. The Lite Reagent contains acridinium-ester-labeled SARS-CoV-2 S1RBD recombinant antigens used to detect anti-SARS-CoV-2 antibodies bound to the SolidPhase.
Follow-up
The members of our team (authors and collaborators) made phone calls to remind participants of collection dates and to deliver results. The telephone calls were made two days before the date of the second dose and on day 28, day 58, day 88, and day 209.
The follow-up times for the AZD1222 vaccine were days 0, 84, 110, 144, and 210 after the first dose; for BNT162b2, the follow-up times were days 0, 21, 60, 90, and 210 after the first dose; and for CoronaVac and Ad26.COV2.S, the follow-up times were days 0, 30, 60, 90, and 210 after the first vaccine dose.
Statistical analysis
The absolute and relative frequencies are presented for the qualitative variables. For the quantitative variables, the results are presented as averages and standard deviations. The assumption of normality was validated using the Shapiro–Wilks test. Subsequently, a comparison between categorical variables was performed using the chi-square test or Fisher's exact test.
To evaluate associations between the independent variables, a multivariate analysis model (Poisson) was performed that considered infection during the study time and seroconversion at 210 days as the dependent variables. Variables with p values <0.2 in the bi-variate analysis were included in the model. We have adjusted the model for covariates to downward bias in estimating the vaccination effect [22]. The most parsimonious model was selected based on the AIC score. The prevalence ratio (PR) was estimated with its respective 95% confidence interval.
The overall crude frequencies of seropositivity tests were estimated. Then, crude seroprevalence was adjusted using the Bayesian method in R V.2.21.2 (pack RStan) using the data of sensitivity and specificity reported in previous studies carried out using CLIA in Colombian populations [23]. 95% Bayesian confidence intervals were obtained. The analyses were performed in R® version 3.6.2.
Comment 3: There are some deficiencies in the experimental design. The author highlighted the importance of exploring more about immunoassays in the introduction section, but no more related experiments were carried out. In addition, the effects of booster doses and unfinished experimenters were not strictly statistically analyzed, which needs to be supplemented. Another basic point is that the authors did not set up a control group that was not vaccinated, which is strange.
Response to reviewer:
We agree with the comment from the reviewer. In order to understand the effect of massive vaccination, immunoassays are needed. Nevertheless, to this point, we don´t count on results related to cellular response and vaccination. To reduce the potential confusion for readers we have modified the introduction and added a paragraph to the limitation section as follows:
Introduction
Vaccination is considered the primary strategy for preventing severe forms of COVID-19 [1]. By March 2022, approximately 64.3% of the world population had received at least one dose of a vaccine against COVID-19 [2]. According to the World Health Organization (WHO), 11,190 million doses have been administered worldwide, and 15.6 million are now administered daily [2]. Evaluating the effectiveness of the COVID-19 vaccines available in the different populations where the vaccines have been administered is necessary, for the generation of knowledge and strategies in both, the current and the future vaccination plans.
Currently, there are approximately 336 studies in progress evaluating different vaccines in different populations and regions of the world [3][4]. The generation of this type of technology, facilitated by greater flexibility in the requirements for approving research in humans, has led to doubts among a large portion of the population [5,6]. Additionally, there is variation in vaccine efficacy for severe forms of the disease [7]. In turn, concerns have been raised about the representation of indigenous and Hispanic populations within clinical trials for vaccines, a lack of which could potentially impact seroconversion outcomes [8,9].
Several countries in the region have provided reports on the efficacy of vaccines used in their populations. For example, Chile reported that the effectiveness of the CoronaVac vaccine was 65.9% (95% CI, 65.2 to 66.6) for the prevention of the disease, 87.5% (95% CI, 86, 7 to 88.2) for hospitalization, and 86.3% (95% CI, 84.5 to 87.9) for avoiding deaths related to COVID-19 [10]. In Uruguay, the effectiveness of the BNT162b2 vaccine was 78% (95% CI 76-79) and for CoronaVac was 59.9% (95% CI 59-60)[11]. In turn, the efficacy concerning preventing deaths was 96% (95% CI 95.3–96.8) for BNT162b2 and 94.65% (95% CI 93.4–95.6) for CoronaVac. Also, a review of the literature reported that full vaccination of COVID-19 vaccines is highly effective in early COVID-19 variants (Alpha, Beta, Gamma, and Delta). Nevertheless, the effectiveness against Delta and Omicron variants is limited [12].
The variation in the studies carried out in the region suggests that there are elements at the individual level that influence the outcomes evaluated.
An additional aspect that has not been addressed in vaccine efficacy studies is the effect they have on the immune system. Although health systems need to understand the effect of vaccination schedules on deaths and transmission, it is also relevant to determine the effect that vaccines have on the immune system at the population level, including humoral and cell responses [13]. This information is generally of interest only for the first phases of clinical trials, but in the context of emerging new strains of interest, with the potential greater transmission or lethality, it is important to fully understand how vaccines stimulate effector cells and immunoglobulin production [14].
Recently, concerns have arisen about the need for administering additional doses or combinations of vaccines to ensure long-term immunity [15]. Knowing at the population level what the effect that vaccination schedules have on immune response, studies focused on antibody response may contribute when making decisions about potential modifications to the National Vaccination Plan. Colombia implemented the National Vaccination Plan against SARS-CoV-2 where vaccination schedules were established for WHO-approved vaccination schedules acquired through multilateral mechanisms (CO-VAX) and by direct negotiation with pharmaceutical companies. The aim of this study was to evaluate the effect of vaccines available in the National Vaccination Plan on the immune system of the Colombian population through the prospective measurement of total antibody markers.
Limitations
Our study has several limitations. First, there were limitations related to the nonprobabilistic consecutive sampling design. Due to the nature of this design, the conclusions are only applicable to this group of participants and cannot be extrapolated to other regions of the country. Furthermore, due to the absence of a control group, it was not possible to compare the outcomes evaluated against the unvaccinated population. As a strength of the study, there is a robust core of studies that have shown the effectiveness of vaccines against COVID-19, compared to placebo, in reducing mortality and ICU hospitalisation [10,32–34].
Third, this study only evaluated the total antibody and IgG responses and did not evaluate neutralizing antibodies and the cellular response against the virus. Recent studies carried out with serum from the Colombian population suggest a reduction in the ability of antibodies to neutralize the variants identified in 2021 compared to the ability of an-tibodies to neutralize the variants identified in 2020 and the original strain identified in 2019 [35].
Fourth, our study did not analyzed the effects on the cellular response. The evaluation of humoral response to vaccines is needed to understand how they contribute to increase the inmme response. Some authors have indicated that the administration of two dosages of BNT162b2 mRNA COVID-19 vaccine increase the expansion of the Ag-specific CD4+CD40L+ T cells [36]. Also, it has been proposed that CoronaVac administration in-creases IFN-gamma- and Granzyme B-producing cells [37].
Finally, although the study recruited a number greater than the minimum sample size, 44% of participants did not complete the study volunteers who received Ad26.COV2 or CoronaVac. This can be explained by two main reasons. The Ad26.COV2 requires only one dose, which facilitated adherence to the study because there was no second “forced” encounter. In turn, for participants who received CoronaVac, the second dose was not administered until day 56 due to the lack of availability of vaccines in the country. With this, the second “forced” encounter was delayed, and in that process, a considerable number of participants were lost. Furthermore, the economic situation in the country forced many of the participants to drop out because they did not have permission from their employers to take time off work to provide samples. However, because we had the identification numbers of the participants, we searched the death records database to determine whether individuals had died.
The referee has noted that we need to evaluate the effect of booster dosage on the experiments. According to this, we have included the booster dosage as an additional variable in the regression model (Table 2). Since the number of participants who have received a booster is reduced the effect of the additional dosage was not significantly associated with a reduction of the COVID-19 incidence (please see section 3.3).
Finally, the reviewer has indicated that we have not used a control group. Indeed, we have designed a prospective cohort of vaccinated participants. This type of epidemiological design doesn´t require a control (unvaccinated group). In addition, there is a piece of enormous evidence that shows that people who receive the vaccination schedule have fewer odds of having severe forms of COVID-19. In this sense, we consider that wasn´t ethical to perform an analysis including a non-vaccinated group.
Reviewer 2 Report
The manuscript of Malagon-Rojas et al. describes the comparison of SARS-CoV-IgG antibody responses after the administration of 4 different biological agents on the Colombian population. The work shows not only the efficacy of the vaccines on the death and transmission prevention but also the effect on the immune system.
Minor comments:
- Line 19: It was used 5 or 4 biological agents?
- Lines 53-55: Effectiveness of what? Is not clear, please rephrase.
- Lines 93-95: This participants with these comorbidities were also enrolled in the study or were excluded. Is not clear this phrase. Please rephrase.
- Please describe the CLIA and RT-PCR method, or give a reference that have the methods described.
- Line 182-183: Please rephrase, the meaning of this phrase is not clear to the reader.
- Figure 2 - Please improve the quality of this figure. Increase the letter size, it is too small.
Author Response
The manuscript of Malagon-Rojas et al. describes the comparison of SARS-CoV-IgG antibody responses after the administration of 4 different biological agents on the Colombian population. The work shows not only the efficacy of the vaccines on the death and transmission prevention but also the effect on the immune system.
Response to reviewer: Many thanks for your valuable feedback. According to your suggestion, we have modified all the lines into the document.
Minor comments:
Comment 1: Line 19: It was used 5 or 4 biological agents?
Response to reviewer: it was modified. We considered four biological agents
Comment 2: Lines 53-55: Effectiveness of what? Is not clear, please rephrase.
Response to reviewer: We have rephrase as follows:
Also, a review of the literature reported that full vaccination of COVID-19 vaccines is highly effective early COVID-19 variants (Alpha, Beta, Gamma, and Delta). Nevertheless, the effectiveness against Delta and Omicron variants is limited [12]
Comment 3: Lines 93-95: This participants with these comorbidities were also enrolled in the study or were excluded. Is not clear this phrase. Please rephrase.
Response to reviewer: We have change this part of the document as follows:
In the analysis were exclude the participants with comorbidities such as diabetes, cardiovascular diseases, hypo/hyperthyroidism, chronic kidney disease, and living with some type of physical disability. Also, participants who, due to time availability, did not perform the scheduled follow-ups and who had medical contraindications were excluded from receiving biologics.
Comment 4: Please describe the CLIA and RT-PCR method, or give a reference that have the methods described.
Response to reviewer: Thank you so much for your suggestion. We have included a CLIA and RT-PCR description as follows:
RNA identification
Nasopharyngeal swab samples were refrigerated and transported to the local labor-atory in each city. The RNA was extracted and amplified according to the Berlin protocol, which was validated Insituto Nacional de Salud [18]. Viral RNA extraction was per-formed by using the QIAamp Viral RNA Mini kit (Qiagen Inc., Chatsworth, CA, USA) or the MagNA Pure LC nucleic acid extraction system (Roche Diagnostics GmbH, Mann-heim, Germany) [18].
Antibodies identification
The ADVIA Centaur COV2 (COV2T & sCOV2G) assay was used in order to measuring total antibodies (IgM+ IgG). The COV2T is an in-sandwich one-step automated antigen test for the qualitative detection of total antibodies (IgM + IgG), against the receptor-binding domain (RBD) of the spike protein of SARS-CoV-2 virus, in serum or plasma. The ADVIA CentaurCOV2 is a fully automated step antigen sandwich immunoassay using acridinium ester chemiluminescent technology, in which antigens are bridged by antibodies present in the sample. The Solid Phase contains a preformed complex of streptavidin-coated micro-particles and biotinylated SARS-CoV-2 spike 1 receptor binding domain (S1 RBD) recombinant antigens. This reagent is used to capture anti-SARS-CoV-2 antibodies in the sample [18]. The Lite Reagent contains acridinium-ester-labeled SARS-CoV-2 S1RBD recombinant antigens used to detect anti-SARS-CoV-2 antibodies bound to the SolidPhase.
Comment 5: Line 182-183: Please rephrase, the meaning of this phrase is not clear to the reader.
Response to reviewer: we modified the phrase as follows:
The infection with COVID-19 of 10.22% of individuals (95% CI 8.25-12.49; n = 81) occurred during the study follow-up. Most of the participants who got the infection during the study had asymptomatic (29.62%) and mild manifestations (70.37%).
Comment 6: Figure 2 - Please improve the quality of this figure. Increase the letter size, it is too small.
Response to reviewer: Thank you for your suggestion. According to this, we modified figure 2.
Reviewer 3 Report
The article entitled “Comparison of SARS-CoV-2 IgG antibody responses genertaed by the administration of Ad26.COV2.S, AZD1222, BNT162b2 or CoronaVac: Longitudinal prospective cohort study in the Colombian population, 2021-2022” focuses on a frequently treated topic by researchers. Here, there are results are confined to Colombian population although the prospective study may have a translational characteristic. The authors report about the missing data on the effect of vaccines on immune system and that both homologous and heterologous boosters cause an increase of T-cell responses. I think that this topic should be a litttle in-depth because it would be of interest for the readers. Only out of curiosity, if the authors have data about the immune system, such as percentage of T-helper CD4+, T-cytotoxic CD8+, and ratio CD4/CD8, it wolud be interesting to add these data that should provide originality. I think that this manuscript is suitable for publication adding general and/or personal data about the response of immune system after anti-COVID vaccine.
Author Response
Comment 1: the article entitled “Comparison of SARS-CoV-2 IgG antibody responses genertaed by the administration of Ad26.COV2.S, AZD1222, BNT162b2 or CoronaVac: Longitudinal prospective cohort study in the Colombian population, 2021-2022” focuses on a frequently treated topic by researchers. Here, there are results are confined to Colombian population although the prospective study may have a translational characteristic. The authors report about the missing data on the effect of vaccines on immune system and that both homologous and heterologous boosters cause an increase of T-cell responses. I think that this topic should be a litttle in-depth because it would be of interest for the readers. Only out of curiosity, if the authors have data about the immune system, such as percentage of T-helper CD4+, T-cytotoxic CD8+, and ratio CD4/CD8, it wolud be interesting to add these data that should provide originality. I think that this manuscript is suitable for publication adding general and/or personal data about the response of immune system after anti-COVID vaccine.
Response to reviewer: We want to thank the reviewer for this valuable comment. Indeed, real-world studies are needed to understand how immune response is induced through large vaccination programs in different regions of the world.
We have mentioned the relevance of cellular and humoral responses in the limitation section. We have eliminated the cellular and humoral response from the abstract and introduction in order to avoid misinterpretations
We understand that the study of T-helper CD4+, T-cytotoxic CD8+, and ratio CD4/CD8 are quite relevant to understand how real-world vaccine administration contribute to reduce the transmission and the development of severe forms of COVID-19.
In spite our study has include the flow cytometry analysis in a sub-sample of the participants (n=30). Unfortunately, those results are in the pipeline but are not ready for this manuscript.
According to your suggestion we have modified the introduction and methods section as follows:
Introduction
Vaccination is considered the primary strategy for preventing severe forms of COVID-19 [1]. By March 2022, approximately 64.3% of the world population had received at least one dose of a vaccine against COVID-19 [2]. According to the World Health Or-ganization (WHO), 11,190 million doses have been administered worldwide, and 15.6 million are now administered daily [2]. Evaluating the effectiveness of the COVID-19 vac-cines available in the different populations where the vaccines have been administered is necessary, for the generation of knowledge and strategies in both, the current and the fu-ture vaccination plans.
Currently, there are approximately 336 studies in progress evaluating different vac-cines in different populations and regions of the world [3][4]. The generation of this type of technology, facilitated by greater flexibility in the requirements for approving research in humans, has led to doubts among a large portion of the population [5,6]. Additionally, there is variation in vaccine efficacy for severe forms of the disease [7]. In turn, concerns have been raised about the representation of indigenous and Hispanic populations within clinical trials for vaccines, a lack of which could potentially impact seroconversion out-comes [8,9].
Several countries in the region have provided reports on the efficacy of vaccines used in their populations. For example, Chile reported that the effectiveness of the CoronaVac vaccine was 65.9% (95% CI, 65.2 to 66.6) for the prevention of the disease, 87.5% (95% CI, 86, 7 to 88.2) for hospitalization, and 86.3% (95% CI, 84.5 to 87.9) for avoiding deaths re-lated to COVID-19 [10]. In Uruguay, the effectiveness of the BNT162b2 vaccine was 78% (95% CI 76-79) and for CoronaVac was 59.9% (95% CI 59-60)[11]. In turn, the efficacy con-cerning preventing deaths was 96% (95% CI 95.3–96.8) for BNT162b2 and 94.65% (95% CI 93.4–95.6) for CoronaVac. Also, a review of the literature reported that full vaccination of COVID-19 vaccines is highly effective in early COVID-19 variants (Alpha, Beta, Gamma, and Delta). Nevertheless, the effectiveness against Delta and Omicron variants is limited [12].
The variation in the studies carried out in the region suggests that there are elements at the individual level that influence the outcomes evaluated.
An additional aspect that has not been addressed in vaccine efficacy studies is the ef-fect they have on the immune system. Although health systems need to understand the effect of vaccination schedules on deaths and transmission, it is also relevant to determine the effect that vaccines have on the immune system at the population level, including humoral and cell responses [13]. This information is generally of interest only for the first phases of clinical trials, but in the context of emerging new strains of interest, with the po-tentially greater transmission or lethality, it is important to fully understand how vaccines stimulate effector cells and immunoglobulin production [14].
Recently, concerns have arisen about the need for administering additional doses or combinations of vaccines to ensure long-term immunity. A better understanding of the ef-fect that vaccination schedules on real world population may contribute when making decisions about potential modifications to the National Vaccination Plan. Colombia im-plemented the National Vaccination Plan against SARS-CoV-2 where vaccination sched-ules were established for WHO-approved vaccination schedules acquired through multi-lateral mechanisms (COVAX) and by direct negotiation with pharmaceutical companies. The aim of this study was to evaluate the effect of vacciness available in the National Vac-cination Plan on the immune system of the Colombian population through the prospec-tive measurement total antibody markers.
- Materials and Methods
Type of study
This was a nonprobabilistic, consecutive cross-sectional study of a Colombian popu-lation who received a vaccination schedule with Ad26.COV2.S (Janssen), AZD1222 (AstraSeneca), BNT162b2 (Pfizer) or CoronaVac through the National Vaccination Plan in the prioritization phase established by the National Government.
Timeline
The National Vaccination Plan established specific criteria for the population and the type of vaccine to be used based on the availability of resources. In this sense, for the study population, the BNT162b2 vaccine was available to the population between March and June 2021; the CoronaVac vaccine was available between May and August 2021; and AZD1222 and Ad26.COV2 were available between June and July 2021.The study was conducted between March 2021 and March 2022.
Eligibility criteria
Men and women between 18 and 80 years old, who were residents of the Colombian territory. Also, voluntaries who at the time of inclusion in the study, intended to receive the complete vaccination schedule for COVID-19 were included.
The participants with these comorbidities were also excluded. We have chante this part of the document as follows:
In the analysis were exclude the participants with comorbidities such as diabetes, cardiovascular diseases, hypo/hyperthyroidism, chronic kidney disease, and living with some type of physical disability. Also, participants who, due to time availability, did not perform the scheduled follow-ups and who had medical contraindications were excluded from receiving biologics.
Participants who, due to time availability, did not perform the scheduled follow-ups and who had medical contraindications were excluded from the study.
Population
The global sample size was estimated for the entire study using as a parameter the percentage of seroconversion generated after the administration of the vaccination sched-ules [16]. The sample size was estimated using OpenEpi v 3.0. A sample size of 589 par-ticipants was estimated, with a confidence level of 95%, a margin of error of 5%, an accu-racy of 2.5%, and an oversample of 30%. Subsequently, the proportional allocation was employed for assigning participants to groups for each vaccine proposed in the National Vaccination Plan [17].
Participants were recruited from five health institutions: Santa Matilde Hospital in Madrid, Santa Matilde Hospital - Bojacá, National Institute of Health, San Andrés Island Departmental Laboratory of Public Health, and San Rafael de Fusagasugá Hospital. After the vaccine was administered, individuals were invited to participate in the study. The people who voluntarily agreed to participate were directed to a site where information was collected.
Sociodemographic characterization and adverse reactions
Each participant responded to a survey designed to collect information regarding health conditions, sociodemographic variables, and symptoms associated with vaccina-tion.
We considered adverse reactions reported by the National Health Services and World Health Organization [18–20]. It was considered mild-moderate symptoms (pain at the in-jection site, fever, fatigue, headache, muscle pain, chills and diarrhea) and severe effects (anaphylaxis, thrombosis with thrombocytopenia syndrome, Guillain-Barré Syndrome, myocaditis, pericarditis and death).
Biological samples collection
A blood sample (7 mL) was collected from each participant by venipuncture using additive-free silicone vacuum tubes containing serum separator gel. Subsequently, the samples were centrifuged at 3,500 rpm for 10 minutes. Then, 2 ml of serum was stored in vials and frozen at -70 °C until processing.
Nasopharyngeal samples were collected using sterile nylon microbiological transport swabs (Improve®, China) in viral transport medium (VTM) (Viral ad-bio®) and frozen at -70 °C until processing.
RNA identification
Nasopharyngeal swab samples were refrigerated and transported to the local labor-atory in each city. The RNA was extracted and amplified according to the Berlin protocol, which was validated Insituto Nacional de Salud [21]. Viral RNA extraction was per-formed by using the QIAamp Viral RNA Mini kit (Qiagen Inc., Chatsworth, CA, USA) or the MagNA Pure LC nucleic acid extraction system (Roche Diagnostics GmbH, Mann-heim, Germany) [21].
Antibodies identification
The ADVIA Centaur COV2 ( COV2T & sCOV2G) assay was used in order to measuring total antibodies (IgM+ IgG). The COV2T is an in-sandwich one-step automated antigen test for the qualitative detection of total antibodies (IgM + IgG), against the receptor-binding domain (RBD) of the spike protein of SARS-CoV-2 virus, in serum or plasma. The ADVIA CentaurCOV2 is a fully automated step antigen sandwich immunoassay using acridinium ester chemiluminescent technology, in which antigens are bridged by antibodies present in the sample. The Solid Phase contains a preformed complex of strept avidin-coated microparticles and biotinylated SARS-CoV-2 spike 1 receptor bindingdomain (S1 RBD) recombinant antigens. This reagent is used to capture anti-SARS-CoV-2 antibodies in the sample [21]. The Lite Reagent contains acridinium-ester-labeled SARS-CoV-2 S1RBD recombinant antigens used to detect anti-SARS-CoV-2 antibodies bound to the SolidPhase.
Follow-up
The members of our team (authors and collaborators) made phone calls to remind participants of collection dates and to deliver results. The telephone calls were made two days before the date of the second dose and on day 28, day 58, day 88, and day 209.
The follow-up times for the AZD1222 vaccine were days 0, 84, 110, 144, and 210 after the first dose; for BNT162b2, the follow-up times were days 0, 21, 60, 90, and 210 after the first dose; and for CoronaVac and Ad26.COV2.S, the follow-up times were days 0, 30, 60, 90, and 210 after the first vaccine dose.
Statistical analysis
The absolute and relative frequencies are presented for the qualitative variables. For the quantitative variables, the results are presented as averages and standard deviations. The assumption of normality was validated using the Shapiro–Wilks test. Subsequently, a comparison between categorical variables was performed using the chi-square test or Fisher's exact test.
To evaluate associations between the independent variables, a multivariate analysis model (Poisson) was performed that considered infection during the study time and sero-conversion at 210 days as the dependent variables. Variables with p values <0.2 in the bi-variate analysis were included in the model. We have adjusted the model for covariates to downward bias in estimating the vaccination effect [22]. The most parsimonious model was selected based on the AIC score. The prevalence ratio (PR) was estimated with its re-spective 95% confidence interval.
The overall crude frequencies of seropositivity tests were estimated. Then, crude se-roprevalence was adjusted using the Bayesian method in R V.2.21.2 (pack RStan) using the data of sensitivity and specificity reported in previous studies carried out using CLIA in Colombian populations [23]. 95% Bayesian confidence intervals were obtained. The analyses were performed in R® version 3.6.2.
Reviewer 4 Report
The article reviewed was describing the results of a sero-survey of vaccinated Columbians during the SARS-CoV-2 pandemic. The main findings of this study include that some vaccines lead to a longer-lived antibody response as measured by CLIA and that older folks tend to lose their antibody response quicker than other cohorts.
I did not find this paper to be novel, but it can be important to survey different areas of the world to get real-time data regarding vaccine efficacy in any population. The authors also mentioned cellular or humoral responses several times, without ever showing data to support themselves.
A major problem with this article is a total lack of information regarding data set "correction" or "adjustment." It is hard to understand why data are being adjusted. There is no data presented in the main body that shows infection or re-infection statistics, even though there is a paragraph in the methods about detecting SARS-CoV-2 RNA.
Another major issue is the way the data are presented. The tables look very scattered, and the graphs are not well labeled or described. In fact the figure legends do nothing to help the reader understand what they are looking at.
A final issue is an enormous lack of detail. In the abstract, the authors claim Columbians were given 5 biological agents. Data for 4 vaccine models are shown. The repeated use of the word "biologics" is really off putting to a reader with virology knowledge. Why not use the word vaccine or immunized? Biologics sounds more appropriate for antibody therapy. Lines 93-97 are not complete sentences. No variables or list or adverse reactions is defined or displayed.
Overall, I think the goal of this type or article is valid and important. However, I think this article should be re-worked from the beginning before being considered for publication.
Author Response
Comment 1: I did not find this paper to be novel, but it can be important to survey different areas of the world to get real-time data regarding vaccine efficacy in any population. The authors also mentioned cellular or humoral responses several times, without ever showing data to support themselves.
Response to reviewer: We want to thank the reviewer for this valuable comment. Indeed, real-world studies are needed to understand how immune response is induced through large vaccination programs in different regions of the world.
We have mentioned the relevance of cellular and humoral responses in the limitation section. We have eliminated the cellular and humoral response from the abstract and introduction in order to avoid misinterpretations
Changes are presented as follows:
Abstract: To mitigate the COVID-19 pandemic by the SARS-CoV-2 coronavirus, vaccines were rapidly developed and introduced in many countries. In Colombia, the population was vaccinated with five biological agents. Therefore, this research aimed to determine the effect of the vaccines introduced in the National Vaccination Plan to prevent SARS-CoV-2 infection, sero-conversion, and the permanence of antibodies in the blood. We conducted a prospective, no probabilistic, and consecutive cross-sectional cohort study in the population with availability to vaccination with CoronaVac, Ad26.COV2.S, AZD1222, and BNT162b2 from March 2021 to March 2022. The study included 1327 vaccinated people. The majority of participants were vaccinated with BNT162b2 (36.1%; n = 480), followed by Ad26.COV2.S (26.9%, n = 358), CoronaVac (24%, n = 331), and AZD1222 (11.9%; n = 158). The crude seroprevalence on day zero varied between 18.1% and 57.8%. Participants who received BNT162b2 had a lower risk of SARS-CoV-2 infection than those who received the other vaccines. Participants who were immunized with BNT162b2 and AZD1222 had a better probability of losing reactivity on day 210 of receiving the vaccine
Third, this study only evaluated the total antibody and IgG responses and did not evaluate neutralizing antibodies and the cellular response against the virus. Recent stud-ies carried out with serum from the Colombian population suggest a reduction in the abil-ity of antibodies to neutralize the variants identified in 2021 compared to the ability of an-tibodies to neutralize the variants identified in 2020 and the original strain identified in 2019 [30].
A major problem with this article is a total lack of information regarding data set "correction" or "adjustment." It is hard to understand why data are being adjusted. There is no data presented in the main body that shows infection or re-infection statistics, even though there is a paragraph in the methods about detecting SARS-CoV-2 RNA.
Comment 2: Another major issue is the way the data are presented. The tables look very scattered, and the graphs are not well labeled or described. In fact the figure legends do nothing to help the reader understand what they are looking at.
Thank the reviewer for pointing this out. We have used the “adjustment” in two parts of the analysis. First, to estimate the risk factor for COVID-19 during the study. Second, to calculate the real-world sensitivity of the CLIA to detect antibodies. Both analyses contribute to reducing bias in the results.
We have modified the statistical analysis section as follows:
Statistical analysis
The absolute and relative frequencies are presented for the qualitative variables. For the quantitative variables, the results are presented as averages and standard deviations. The assumption of normality was validated using the Shapiro–Wilks test. Subsequently, a comparison between categorical variables was performed using the chi-square test or Fisher's exact test.
To evaluate associations between the independent variables, a multivariate analysis model (Poisson) was performed that considered infection during the study time and sero-conversion at 210 days as the dependent variables. Variables with p values <0.2 in the bi-variate analysis were included in the model. We have adjusted the model for covariates to a downward bias in estimating the vaccination effect [23]. The most parsimonious model was selected based on the AIC score. The prevalence ratio (PR) was estimated with its re-spective 95% confidence interval.
The overall crude frequencies of seropositivity tests were estimated. Then, crude se-roprevalence was adjusted using the Bayesian method in R V.2.21.2 (pack RStan) using the data of sensitivity and specificity reported in previous studies carried out using CLIA in Colombian populations [22]. 95% Bayesian confidence intervals were obtained. The analyses were performed in R® version 3.6.2.
Also, we have modified the legend of the figure 2 as follows:
The figure shows the proportion of participants who were reactive during the follow-up time (crude in blue and adjusted seroprevalence in red). The crude proportion shows the part of the participants who were reactive. In order to determiinethe real proportion of seroreactivity, an adjustment was performed based on the sensitivity and specificity of the ADVIA Centaur COV2, previously reported [22]. The figure also presents the prevalence of total antibodies against COVID-19 during the study (day 0,30,60, 90 and 2010). Participants who received BNT162b2 and AZD1222 had greater seroreactivity ratios than those who received Ad26.COV2 and CoronaVac. However, by day 210 of follow-up, these reactivity ratios were greater than 70% for the four vaccines.
Comment 3: A final issue is an enormous lack of detail. In the abstract, the authors claim Columbians were given 5 biological agents. Data for 4 vaccine models are shown. The repeated use of the word "biologics" is really off putting to a reader with virology knowledge. Why not use the word vaccine or immunized? Biologics sounds more appropriate for antibody therapy.
Response to reviewer: Thank you so much for your observation. We modified the abstract, also, we are so sorry about the confusion about the number of vaccines. We have corrected this by four agents.
Also, we want to apologize for the wrong use of the word “biologic”. According to your suggestion, we have eliminated the word “biologics” from the whole manuscript.
Comment 4: lines 93-97 are not complete sentences. No variables or list or adverse reactions is defined or displayed.
Response to reviewer: Thank you very much, we agree with your comment. We have modified it as follows:
In the analysis were exclude the participants with comorbidities such as diabetes, cardiovascular diseases, hypo/hyperthyroidism, chronic kidney disease, and living with some type of physical disability. Also, participants who, due to time availability, did not perform the scheduled follow-ups and who had medical contraindications were excluded from receiving biologics. Each participant responded to a survey designed to collect information regarding health conditions, sociodemographic variables, and symptoms associated with vaccination.
We considered adverse reactions reported by the National Health Services and World Health Organization [18–20]. It was considered mild-moderate symptoms (pain at the in-jection site, fever, fatigue, headache, muscle pain, chills and diarrhea) and severe effects (anaphylaxis, thrombosis with thrombocytopenia syndrome, Guillain-Barré Syndrome, myocaditis, pericarditis and death).
Comment 5: Overall, I think the goal of this type or article is valid and important. However, I think this article should be re-worked from the beginning before being considered for publication.
Response to reviewer: Thank you for giving us the opportunity to rework this document following your contributions as an expert. We hope it can be useful to the knowledge of this terrible pandemic that we are going through, and understand the aspects of prevention strategies from underdeveloped countries such as Colombia.
According to your suggestion, we have carefully reviewed the full manuscript, performing some changes in the introduction, methods, discussion and conclusion sections. Please, see the track version of the manuscript that shows the highlighted changes (yellow).
Round 2
Reviewer 3 Report
I read the revised version of the article entitled “Comparison od SARS-CoV-2 IgG antibody responses generated by the administration of Ad26.COV2.S, AZD1222, , BNT162b2 or Corona Vac: Longitudinal prospective cohort study in the Colombian population, 2021-2022”. The authors modified the text in according to the comments of the reviewer. Therefore, I think that this manuscript is suitable for publication.
Author Response
Many thanks to the reviewer for his comments
Reviewer 4 Report
The authors have sufficiently addressed my concerns. The data tables need to be re-formatted to ensure ease of viewing, such as the addition of lines to separate those with a CoV infection and those without, but these changes are stylistic and don't affect the overall interpretations.
Author Response
Many thanks to the reviewers for this suggestion. According to your comments, we have used the services of an English editor who has checked the manuscript. Also, we have performed the recommended changes in Tables 1 and 2.
Regards,